# SOSP: Efficiently Capturing Global Correlations by Second-Order Structured Pruning

## Abstract

Pruning neural networks reduces inference time and memory cost, as well as accelerates training when done at initialization. On standard hardware, these benefits will be especially prominent if coarse-grained structures, like feature maps, are pruned. We devise global saliency-based methods for second-order structured pruning (SOSP) which include correlations among structures, whereas highest efficiency is achieved by saliency approximations using fast Hessian-vector products. We achieve state-of-the-art results for various object classification benchmarks, especially for large pruning rates highly relevant for resource-constrained applications. We showcase that our approach scales to large-scale vision tasks, even though it captures correlations across all layers of the network. Further, we highlight two outstanding features of our methods. First, to reduce training costs our pruning objectives can also be applied at initialization with no or only minor degradation in accuracy compared to pruning after pretraining. Second, our structured pruning methods allow to reveal architectural bottlenecks, which we remove to further increase the accuracy of the networks.

## 1 Introduction

Deep neural networks have consistently grown in size over the last years with increasing performance. However, this increase in size leads to slower inference, higher computational requirements and higher cost. To reduce the size of the networks without affecting their performance, a large number of pruning algorithms have been proposed (e.g., LeCun et al., 1990; Hassibi et al., 1993; Reed, 1993; Han et al., 2015; Blalock et al., 2020). Pruning can either be *unstructured*, i.e. removing individual weights, or *structured*, i.e. removing entire substructures like nodes or channels. Single-shot pruning methods, as investigated in this work, usually consist of three steps: 1) training, 2) pruning, 3) another training step often referred to as *fine-tuning*.

Unstructured pruning can significantly reduce the number of parameters of a neural network with only little loss in the accuracy, but the resulting networks often show only a marginal improvement in training and inference time, unless specialized hardware is used (He et al., 2017). In contrast, structured pruning can directly reduce inference time and even training time when applied at initialization (Lee et al., 2018). To exploit these advantages, in this work, we focus on structured pruning.

Most sensitivity-based pruning methods such as OBD (e.g., LeCun et al., 1990) or C-OBD (Wang et al., 2019a) evaluate the effect of removing a single weight or structure on the loss of the neural network, while neglecting possible correlations between different structures and within the structures themselves. This can significantly harm the estimation of the sensitivities. We take these correlations into account by applying efficient second-order estimations that not only consider the diagonal terms of the Hessian, but also all off-diagonal terms.

Global pruning removes structure by structure from all available structures of a network until a predefined percentage of pruned structures is reached. Recent examples for global structured pruning methods are NN Slimming (Liu et al., 2017), C-OBD and EigenGamage (Wang et al., 2019a). Local pruning, on the other hand, first subdivides all global structures into subsets (e.g. layers) and removes a percentage of structures of each subset. Recent examples for local pruning methods are HRank (Lin et al., 2019), CCP (Peng et al., 2019), FPGM (He et al., 2019) and Variational Pruning (Zhao et al., 2019). Most local pruning schemes use a predefined layer-wise pruning ratio, which fixes the percentage of structures removed per layer. While this approach prevents the layers from collapsing, it also reduces some of the degrees of freedom, since some layers may be less important than others.

Our main goal in this work is to devise a simple and efficient second-order pruning method, which considers all global correlations for structured sensitivity pruning. In addition, we want to highlight the benefits that such methods may have over other structured global and local pruning schemes.

Our contributions are as follows:

- We introduce two novel saliency-based methods for second-order structured pruning (SOSP), which consider all correlations across structures and layers. We benchmark our SOSP methods against a variety of state-of-the-art pruning methods on several networks and datasets and achieve comparable or better results at low computational costs.

- We show that our pruning methods can also be applied at initialization almost matching the performance of pruning after training and significantly reducing the cost of network training.

- We exploit the structure of the pruning masks found by our SOSP methods to remove architectural bottlenecks, which further improves the performance of the pruned networks. In this work, we consider layers with disproportionally low pruning ratios architectural bottlenecks.

PyTorch code implementing our method is attached in the Supplementary Material and we will publish the code upon acceptance of this manuscript.

## 2 SOSP: Second-order structured pruning

A neural network (NN) maps an input $x \in \mathbb{R}^d$ to an output $f_\theta(x) \in \mathbb{R}^D$, where $\theta \in \mathbb{R}^P$ are its $P$ parameters. NN training proceeds, after random initialization $\theta = \theta_0$ of the weights, by mini-batch stochastic gradient descent on the empirical loss $\mathcal{L}(\theta) := \frac{1}{N} \sum_{n=1}^{N} \ell(f_\theta(x_n), y_n)$, given the training dataset $\{(x_1, y_1), \ldots, (x_N, y_N)\}$. In the classification case, $y \in \{1, \ldots, D\}$ is a discrete ground-truth label and $\ell(f_\theta(x), y) := -\log \sigma(f_\theta(x))_y$ the cross-entropy loss, with $\sigma : \mathbb{R}^D \to \mathbb{R}^D$ the softmax-function. For regression, $y \in \mathbb{R}^D$ and $\ell(f_\theta(x), y) = \frac{1}{2} \|f_\theta(x) - y\|^2$ is the squared loss.

Structured pruning aims to remove weights or rather entire structures from a NN $f_\theta$ with parameters $\theta$. A structure can be a filter (channel) in a convolutional layer, a neuron in a fully-connected layer, or an entire layer in a parallel architecture. We assume the NN in question has been segmented into $S$ structures $s = 1, \ldots, S$, which can potentially be pruned. We define the notation $\theta_s \in \mathbb{R}^P$ as the vector whose only nonzero components are those weights from $\theta$ that belong to structure $s$.[1] Then, a *pruning mask* is a set $M = \{s_1, \ldots, s_m\}$ of structures. Applying a mask $M$ to a NN $f_\theta$ means to consider the NN with parameter vector $\theta_{\setminus M} := \theta - \sum_{s \in M} \theta_s$.[2]

We now develop our pruning methods that incorporate global correlations into their saliency assessment by efficiently including the second-order loss terms. The first method (SOSP-I) admits a direct interpretation in terms of individual loss sensitivities, while the second (SOSP-H) remains very efficient for the largest networks due to its Hessian-vector product approximation.

The basic idea behind both our pruning methods is to select the pruning mask $M$ so as to (approximately) minimize the *joint* effect on the network loss

$$\lambda(M) := \left| \mathcal{L}(\theta) - \mathcal{L}(\theta_{\setminus M}) \right|$$

---

[1] We require that each weight is assigned to at most one structure. In practice, we associate with each structure those weights that go *into* the structure, rather than those that leave it.

[2] Note, a mask $M$ learned on $\theta$ can be applied to a different $\theta' \neq \theta$ on the same NN architecture.

of removing all structures in $M$, subject to a constraint on the overall pruning ratio. To circumvent this exponentially large search space, we approximate the loss up to second order, so that

$$\lambda_2(M) = \left| \sum_{s \in M} \theta_s^T \frac{d\mathcal{L}(\theta)}{d\theta} - \frac{1}{2} \sum_{s,s' \in M} \theta_s^T \frac{d^2\mathcal{L}(\theta)}{d\theta \, d\theta^T} \theta_{s'} \right| \tag{1}$$

collapses to single-structure contributions plus pairwise correlations; note that the latter include interactions among the weights within a single $s = s'$, which can be sizeable for large structures.

The first-order terms $\lambda_1(s) := \theta_s \cdot d\mathcal{L}(\theta)/d\theta \in \mathbb{R}^P$ in (1) are efficient to evaluate by computing the gradient $d\mathcal{L}(\theta)/d\theta \in \mathbb{R}^P$ once and then a (sparse) dot product for every $s$. In contrast to these first-order terms, the network Hessian $H(\theta) := d^2\mathcal{L}(\theta)/d\theta^2 \in \mathbb{R}^{P \times P}$ in (1) is prohibitively expensive to compute or store in full. We therefore propose two different schemes to efficiently overcome this obstacle. Each scheme entails its own way to select the pruning mask. We name the full methods SOSP-I (individual sensitivities) and SOSP-H (Hessian-vector product).

## 2.1 SOSP-I: Saliency from individual sensitivities

SOSP-I approximates each individual term $\theta_s^T H(\theta) \theta_{s'}$ in (1) efficiently, as we will show in Eq. (6). We can therefore consider a modification of Eq. (1) in which the sensitivity is judged by considering all single and pairwise sensitivities individually:

$$\lambda_2^I(M) = \sum_{s \in M} \left| \theta_s^T \frac{d\mathcal{L}(\theta)}{d\theta} \right| + \frac{1}{2} \sum_{s,s' \in M} \left| \theta_s^T H(\theta) \theta_{s'} \right|. \tag{2}$$

To avoid cancellations between signed contributions, we take absolute values because this measures the strengths of the individual sensitivities $\lambda_1(s)$ and pairwise correlations $\theta_s^T H(\theta) \theta_{s'}$. While objectives other than $\lambda_2^I$ are equally possible in the method, including $\lambda_2$ and modifications with the absolute value not pulled in all the way, we found empirically that $\lambda_2^I$ performs best overall.

Then, SOSP-I iteratively selects the structures to prune, based on the objective (2): Starting from an empty pruning mask $M = \{\}$, we iteratively add to $M$ the structure $s \notin M$ that minimizes the overall sensitivity $\lambda_2^I(M \cup \{s\})$. In practice, the algorithm pre-computes the matrix $Q \in \mathbb{R}^{S \times S}$,

$$Q_{s,s'} := \frac{1}{2} \left| \theta_s^T H(\theta) \theta_{s'} \right| + \left| \theta_s^T \frac{d\mathcal{L}(\theta)}{d\theta} \right| \cdot \delta_{s=s'}, \tag{3}$$

and selects at each iteration a structure $s \notin M$ to prune by

$$\arg\min_{s \notin M} \lambda_2^I(M \cup s) - \lambda_2^I(M) = \arg\min_{s \notin M} \left( Q_{s,s} + 2 \sum_{s' \in M} Q_{s,s'} \right), \tag{4}$$

terminating at the desired pruning ratio.

In order to approximate the Hessian terms $\theta_s^T H(\theta) \theta_{s'}$ efficiently, we omit from $H(\theta) = \frac{1}{N} \sum_n \nabla_\theta^2 \ell(f_\theta(x_n), y_n)$ those terms that involve the expensive second-order derivatives $\nabla_\theta^2 f_\theta(x_n)$ of the NN outputs, while including second-order couplings due to $\ell$. This is equivalent to approximating $H(\theta) \approx H(f_\theta^{lin}) := \frac{1}{N} \sum_n \nabla_\theta^2 \ell(f_\theta^{lin}(x_n), y_n)$ for the linearized $f_{\theta'}(x) \approx f_{\theta'}^{lin}(x) := f_\theta(x) + \phi(x) \cdot (\theta' - \theta)$ with $\phi(x) := \nabla_\theta f_\theta(x) \in \mathbb{R}^{D \times P}$, which is well motivated by the NTK limit (Jacot et al., 2018) for large NNs at both initialization and after training. The terms in the sum become

$$\nabla_\theta^2 \ell \left( f_\theta^{lin}(x_n), y_n \right) = \phi(x_n)^T R_n \phi(x_n), \tag{5}$$

where $R_n \in \mathbb{R}^{D \times D}$ is diagonal for squared loss, and has an additional rank-1 contribution for cross-entropy (see App. B). Similar Hessian approximations were employed before in NNs (Hassibi et al., 1993; Wang et al., 2019a; Peng et al., 2019) and also in the Gauss-Newton optimization method (Fletcher, 2013). Our final approximation is to use a random subsample of $N' < N$ data points:

$$\theta_s^T H(\theta) \theta_{s'} \approx \frac{1}{N'} \sum_{n=1}^{N'} (\phi(x_n)\theta_s)^T R_n (\phi(x_n)\theta_{s'}). \tag{6}$$

In practice, one pre-computes all (sparse) products $\phi(x_n)\theta_s \in \mathbb{R}^D$ starting from the efficient gradient $\phi(x_n)$, before aggregating a batch onto the terms $\theta_s^T H(\theta) \theta_{s'}$. Eq. (6) also has an interpretation as output correlations between certain network modifications, without using derivatives (App. C).

## 2.2 SOSP-H: Saliency from Hessian-vector product

SOSP-H treats the second-order terms in (1) in a way that is motivated by the limit of large pruning ratios: At high pruning ratios, the sum $\sum_{s' \in M} \theta_{s'}$ in (1) can be approximated by $\sum_{s'=1}^{S} \theta_{s'} =: \theta_{struc}$ (this equals $\theta$ if every NN weight belongs to some structure $s$). The second-order term $\sum_{s,s' \in M} \theta_s^T H(\theta) \theta_{s'} \approx \left( \sum_{s \in M} \theta_s^T \right) \left( H(\theta) \theta_{struc} \right)$ thus becomes tractable since the Hessian-vector product $H(\theta) \theta_{struc}$ is efficiently computable by a variant of the backpropagation algorithm. To account for each structure $s$ and for the first- and second-order contributions separately, as above, we place absolute value signes in (1) so as to arrive at the final objective $\lambda_2^H(M) := \sum_{s \in M} \lambda_2^H(s)$ with

$$\lambda_2^H(s) := \left| \theta_s^T \frac{d\mathcal{L}(\theta)}{d\theta} \right| + \frac{1}{2} \left| \theta_s^T \left( H(\theta) \theta_{struc} \right) \right|. \tag{7}$$

The last term measures the correlations between one structure $s$ and all other prunable structures, although some of these may cancel unlike for SOSP-I. To minimize $\lambda_2^H(M)$, SOSP-H starts from an empty pruning mask $M = \{\}$, and successively adds to $M$ a structure $s \notin M$ with smallest $\lambda_2^H(s)$.

Unlike the Gauss-Newton approximation in SOSP-I, SOSP-H uses the exact Hessian $H(\theta)$, but can therefore not account for individual absolute $s$-$s'$-correlations, see Eq. (7) vs. (2). Both methods reduce to the same first-order pruning method when neglecting the second order (i.e. $H(\theta) := 0$).

## 2.3 Computational complexity

We detail here the computational complexities of our methods (for the experimental evaluation see Sec. 3.2). The approximation of $Q$ in (3) requires complexity $O\left(N'D(F+P)\right) = O(N'DF)$ for computing all $\phi(x_n)\theta_s$, where $F \geq P$ denotes the cost of one forward pass through the network ($F \approx P$ for fully-connected NNs), plus $O(N'DS^2)$ for the sum in (6). This is tractable for modern NNs, while including the exact $H(\theta)$ would have complexity at least $O(N'DSF)$. Once $Q$ has been computed, the selection procedure based on (4) has overall complexity $O(S^3)$, which is feasible for most modern convolutional NNs (Sec. 3.2). The total complexity of the SOSP-I method is thus

$$O(N'DF) + O(N'DS^2) + O(S^3). \tag{8}$$

SOSP-H has computational complexity $O(N'DF)$ to compute the sensitivities (7), which is comparable to computing the sensitivities $Q$ in SOSP-I when the number of structures is low ($S^2 \lesssim F$). Together with the sorting of the saliency values $\lambda_2^H(s)$, the overall complexity of SOSP-H is thus

$$O(N'DF) + O(S \log(S)). \tag{9}$$

Due to its weak dependency on $S$, in practice, SOSP-H efficiently scales to large modern networks and may even be used for *unstructured* second-order pruning, where $S = P$.

Both of our methods scale much better than naively including all off-diagonal Hessian terms, which is intractable for modern NNs due to its $O(N'DSF)$ scaling. Since SOSP-I builds on individual absolute sensitivities and the established Gauss-Newton approximation, we use SOSP-I in the following in particular to validate the more efficient SOSP-H method.

# 3 Results

To evaluate our methods, we train and prune VGGs (Simonyan & Zisserman, 2014), ResNets (He et al., 2016), and DenseNets (Huang et al., 2017) on the Cifar10/100 (Krizhevsky et al., 2009) and ImageNet (Deng et al., 2009) datasets. Stochastic gradient descent with an initial learning rate of 0.1, a momentum of 0.9 and weight decay of $10^{-4}$ is used to train these networks. For ResNet-32/56 and VGG-Net on Cifar10/100, we use a batch size of 128, train for 200 epochs and reduce the learning rate by a factor of 10 after 120 and 160 epochs. To fine-tune the network after pruning, we exactly repeat this learning rate schedule. For DenseNet-40 on Cifar10/100, we train for 300 epochs and reduce the learning rate after 150 and 225 epochs. For ResNets on ImageNet, we use a batch size of 256, train for 128 epochs and use a cosine learning rate decay. For all networks, we prune feature maps (i.e. channels) from all layers except the last fully-connected layer; for ResNets, we also exclude the downsampling-path from pruning. We approximate the Hessians by a subsample of size $N' = 1000$ (see Sec. 2.1). We report the best or average final test accuracy over 3 trials if not noted otherwise. The experiments were run on an internal cluster with Nvidia Tesla V100 GPUs. Reproducing the results presented in this paper would take about 60 days of GPU run-time.

Table 1: Comparison of SOSP to other global pruning methods for high pruning ratios. The comparison for moderate pruning ratios is defered to the appendix (see App. A.1). We tuned our pruning ratios to similar values as reported by the referred methods. To ensure identical implementations of the network models in PyTorch, reference numbers are taken from Wang et al. (2019a) and (Mingjie & Zhuang, 2018). In accordance with all referred methods, we report the mean and standard deviation of the best accuracies observed during fine-tuning. For final accuracies after fine-tuning see App. A.3. * denotes the baseline model. Both SOSP methods perform either on par or outperform the competing global pruning methods.

| Dataset | **Cifar10** | | | **Cifar100** | | |
|---|---|---|---|---|---|---|
| Method | Test acc. (%) | Reduct. in weights (%) | Reduct. in MACs (%) | Test acc. (%) | Reduct. in weights (%) | Reduct. in MACs (%) |
| **VGG-Net*** | 94.18 | - | - | 73.45 | - | - |
| NN Slimming | 85.01 | 97.85 | 97.89 | 58.69 | 97.76 | 94.09 |
| NN Slim. $+L_1$ | 91.99 | 97.93 | 86.00 | 57.07 | 97.59 | 93.86 |
| C-OBD | $92.34 \pm 0.18$ | $97.68 \pm 0.02$ | $77.39 \pm 0.36$ | $58.07 \pm 0.60$ | $97.97 \pm 0.04$ | $77.55 \pm 0.25$ |
| EigenDamage | $92.29 \pm 0.21$ | $97.15 \pm 0.04$ | $86.51 \pm 0.26$ | $\mathbf{65.18} \pm 0.10$ | $97.31 \pm 0.01$ | $88.63 \pm 0.12$ |
| SOSP-I (ours) | $92.62 \pm 0.14$ | $97.79 \pm 0.02$ | $83.52 \pm 0.29$ | $64.20 \pm 0.23$ | $97.83 \pm 0.04$ | $87.02 \pm 0.20$ |
| SOSP-H (ours) | $\mathbf{92.71} \pm 0.19$ | $97.81 \pm 0.01$ | $86.32 \pm 0.29$ | $64.59 \pm 0.35$ | $97.81 \pm 0.01$ | $86.32 \pm 0.29$ |
| **ResNet-32*** | 95.30 | - | - | 76.8 | - | - |
| C-OBD | $91.75 \pm 0.42$ | $97.30 \pm 0.06$ | $93.50 \pm 0.37$ | $59.52 \pm 0.24$ | $97.74 \pm 0.08$ | $94.88 \pm 0.08$ |
| EigenDamage | $\mathbf{93.05} \pm 0.23$ | $96.05 \pm 0.03$ | $94.74 \pm 0.02$ | $65.72 \pm 0.04$ | $95.21 \pm 0.04$ | $94.62 \pm 0.06$ |
| SOSP-I (ours) | $92.43 \pm 0.09$ | $95.47 \pm 0.33$ | $94.07 \pm 0.66$ | $67.36 \pm 0.46$ | $92.69 \pm 0.07$ | $95.63 \pm 0.13$ |
| SOSP-H (ours) | $92.23 \pm 0.12$ | $95.26 \pm 0.10$ | $94.45 \pm 0.40$ | $\mathbf{68.42} \pm 0.21$ | $94.08 \pm 0.21$ | $95.06 \pm 0.14$ |
| **DenseNet-40*** | 94.58 | - | - | 74.11 | - | - |
| NN Slim. $+ L_1$ | 94.22 | 54.21 | - | $\mathbf{73.19}$ | 54.21 | - |
| SOSP-I (ours) | $94.21 \pm 0.04$ | $47.00 \pm 0.10$ | $36.35 \pm 0.12$ | $73.05 \pm 0.11$ | $45.22 \pm 0.10$ | $42.05 \pm 1.16$ |
| SOSP-H (ours) | $\mathbf{94.23} \pm 0.05$ | $49.39 \pm 0.65$ | $38.86 \pm 0.70$ | $73.05 \pm 0.24$ | $48.58 \pm 0.22$ | $42.05 \pm 0.35$ |

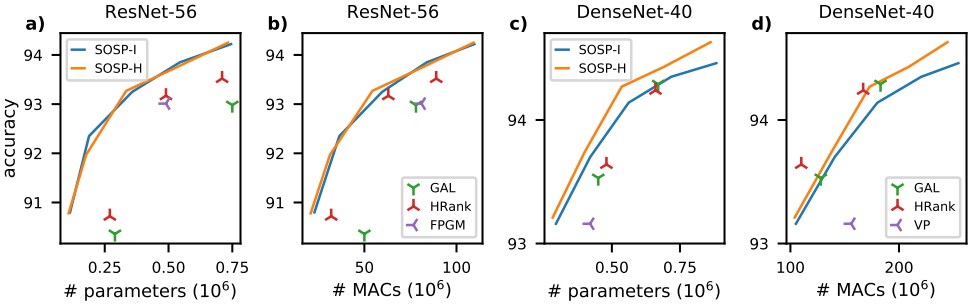

Figure 1: Comparison of SOSP to local, i.e. layer-wise, pruning methods on Cifar10. The best final test accuracy is plotted over the effective number of model parameters (a, c) and MACs (b, d). A tabular representation as well as statistics across trials are shown in App. A.4. SOSP outperforms all competing layer-wise pruning methods, especially over the number of effective parameters.

### 3.1 Comparison to Literature

Here we benchmark the performance of our SOSP methods against existing pruning algorithms on different datasets and networks. First, we compare against other recent global pruning methods, then against local structured pruning methods, i.e. with pre-specified layer-wise pruning rates. In all comparisons we report the achieved test accuracy, the number of parameters of the pruned network and the MACs (often referred to as FLOPs). To facilitate direct comparisons, we report test accuracies in the same way as the competing methods (e.g. best trial or average over trials), but additionally report mean and standard deviation of the test error for our models in App. A. Our count of the network parameters and MACs is based on the actual pruned network architecture (cf. App. D), even though our saliency measure associates with each structure only the weights *into* this structure.

We first compare our SOSP methods with global pruning methods on VGG-Net, ResNet-32 and DenseNet-40. We use the same variants and implementations of these networks as used by Neural Network Slimming (NN Slimming; Liu et al., 2017) as well as EigenDamage and C-OBD (Wang et al., 2019a), e.g. capping the layer-wise ratio of removed structures at 95% for VGGs to prevent layer collapse and increasing the width of ResNet-32 by a factor 4. C-OBD is a structured variant of

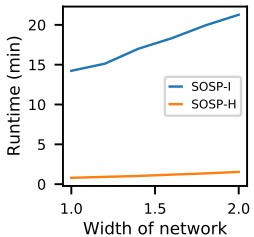

Figure 2: Runtime to calculate the pruning masks for ResNet-56 on Cifar10 over the width of the network for SOSP-I and SOSP-H. We vary the width of the network by increasing the width of each layer by a multiplicative factor.

Table 2: Best final test accuracies and pruning ratios (PR) across 2 trials on ImageNet. For comparison to CCP, we also provide their alternative MAC count (for details, see App. D). * denotes SOSP with kernel scaling (see main text). SOSP outperforms all three competing methods.

| Model | Top-1% (Gap) | Parameters (PR) | MACs (PR) | Alt. MACs (PR) |
|---|---|---|---|---|
| **ResNet-18** | 69.76 (0.0) | 11.7M (0%) | 1.82B (0%) | 1.82B (0%) |
| SOSP (ours) | 69.63 (0.13) | 7.12M (39%) | 1.37B (24%) | 1.31B (28%) |
| FPGM | 68.41 (1.87) | 7.10M (39%) | 1.06B (41%) | - |
| SOSP (ours) | 68.78 (0.98) | 6.42M (45%) | 1.29B (29%) | 1.20B (34%) |
| **ResNet-50** | 76.15 (0.0) | 25.5M (0%) | 3.85B (0%) | 3.85B (0%) |
| SOSP (ours) | 76.56(-0.41) | 19.9M(22%) | 3.06B(21%) | 2.72B(29%) |
| SOSP* (ours) | 76.60 (-0.45) | 17.9M (30%) | 2.79B (28%) | 2.47B (36%) |
| HRank | 74.98 (1.17) | 16.2M (36%) | 2.30B (44%) | - |
| FPGM | 75.59 (0.56) | 15.9M (37%) | 2.36B (42%) | - |
| SOSP (ours) | 75.85 (0.30) | 15.4M (40%) | 2.44B (27%) | 1.97B (49%) |
| HRank | 71.98 (4.17) | 13.8M (46%) | 1.55B (62%) | - |
| CCP | 75.21 (0.94) | - | - | 1.77B (54%) |
| SOSP* (ours) | 75.21 (0.94) | 13.0M (49%) | 2.13B (45%) | 1.68B (56%) |
| SOSP (ours) | 74.39 (1.76) | 11.8M (54%) | 1.89B (51%) | 1.38B (64%) |
| SOSP* (ours) | 73.38 (2.77) | 9.9M (61%) | 1.58B (59%) | 1.10B (72%) |

the original OBD algorithm (Hassibi et al., 1993), which neglects all cross-structure correlations that, in contrast, SOSP takes into account. The results over three trials for high pruning ratios are shown in Tab. 1 and for moderate pruning ratios in App. A.1. To enable the comparison to NN Slimming on an already pretrained VGG, we included the results of NN Slimming applied to a baseline network obtained without modifications to its initial training, i.e. without $L_1$-regularization on the batch-normalization parameters. For moderate pruning rates, all pruning schemes approximately retain the baseline performance for VGG-Net and ResNet-32 on Cifar10 and VGG-Net on Cifar100 (see Tab. 3). The only exception is the accuracy for C-OBD applied to VGG-Net on Cifar100, which drops by approximately $1\%$. For ResNet-32 on Cifar100 the accuracy after pruning is approximately $1\%$ lower than the baseline, for all pruning schemes. In the regime of larger pruning ratios of approximately $97\%$, SOSP and EigenDamage significantly outperform NN Slimming and C-OBD. SOSP performs on par with EigenDamage, except for ResNet-32 on Cifar100, where SOSP outperforms EigenDamage by almost $3\%$. This result indicates that SOSP outperforms all other methods especially in the regime of baseline networks that have relatively few parameters already and on difficult datasets most relevant for applications. For DenseNet-40, we achieve similar results compared to NN Slimming. However, note that NN Slimming requires the modification of network pretraining.

Next, we compare our SOSP methods against four recently published local, i.e. layer-wise, pruning algorithms: FPGM He et al. (2019), GAL (Lin et al., 2019), CCP (Peng et al., 2019), Variational Pruning (VP; Zhao et al., 2019) and HRank (Lin et al., 2020). For ResNet-56, our SOSP methods outperform all other methods across all pruning ratios (see Fig. 1a and b). For DenseNet-40, SOSP achieves better accuracies when compared over parameters (Fig. 1c) and is on par with the best other methods over MACs (Fig. 1d). The reason for this discrepancy is probably that the SOSP objective is agnostic to the number of MACs (image size) in each individual structure.

### 3.2 Scalability and application to large-scale datasets

Before going to large datasets, we compare the scalability of our methods SOSP-I and SOSP-H. As the preceding section shows, both methods perform basically on par with each other in terms of accuracy. This confirms that SOSP-H is not degraded by the approximations leading to the efficient Hessian-vector product, or is helped by use of the exact Hessian. In terms of efficiency, however, SOSP-H shows clear advantages compared to SOSP-I, for which the algorithm to select the structures to be pruned scales with $O(S^3)$ (see Sec. 2.3) potentially dominating the overall runtime for large-scale networks. Measurements of the actual runtimes show that already for medium-sized networks SOSP-H is more efficient than SOSP-I (Fig. 2). Since SOSP-I becomes impractical for large-scale networks, for the ImageNet dataset we will only evaluate SOSP-H and refer to it as SOSP.

On ImageNet, we compare our results to literature for ResNet-18 and ResNet-50, see Tab. 2. Because SOSP assesses the sensitivity of each structure independently of the contributed MACs, it has a bias towards pruning small-scale structures. This tendency is strongest for ResNet-50, due to its $1 \times 1$-convolutions. Since these $1 \times 1$-convolutions tend to contribute disproportionally little to the

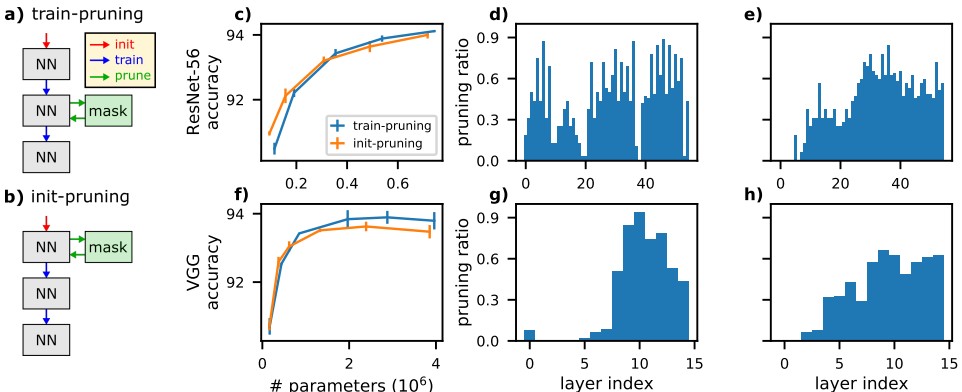

Figure 3: Comparison between pruning after training and at initialization on Cifar10. Both pruning schemes, train-pruning (a) and init-pruning (b), train the network for the same overall number of epochs, but generate and apply the pruning masks at different point in times. The average and standard deviation of the test accuracy across 3 trials is plotted against the number of model parameters for ResNet-56 (top row; c) and VGG (bottom row; f). For a single trial, in which overall $50\%$ of the structures are pruned, we visualize the pruning masks of train-pruning and init-pruning by showing the layer-wise pruning ratios in (d, g) and (e, h), respectively.

overall number of MACs, we devised a scaled variant of SOSP, which divides the saliency of every structure by the kernel size (e.g. 1, 3 or 7). Compared to the vanilla SOSP, the scaled variant of SOSP is able to remove larger percentages of MACs with similar drops in accuracy (see Tab. 2).

For both networks SOSP outperforms HRank and FPGM, especially when considering the main objective of SOSP, which is to reduce the number of parameters or structures. Since CCP uses a different way of calculating the MACs, which leads to consistently higher pruning ratios, we added an alternative MAC count to enable a fair comparison (for details, see App. D). Since HRank and FPGM do not mention their MAC counting convention, we assume they use the same convention as we do. Taking this into account, our scaled SOSP variant is able to prune more MACs than CCP, while having the same final accuracy.

### 3.3 Pruning at Initialization

Traditionally, pruning methods are applied to pretrained networks, as also done in the previous sections, but recently there has been growing attention on pruning at initialization following the works of Lee et al. (2018) and Frankle & Carbin (2018). Since SOSP employs the absolute value of the sensitivities, it can also be applied to a randomly initialized network without any modifications. Thus, SOSP can also be seen as an efficient second-order generalization of SNIP (Lee et al., 2018; van Amersfoort et al., 2020). While EigenDamage can in principle be modified and applied to a randomly initialized network, NN Slimming can not be applied at initialization.

Usually pruning at initialization leads to worse accuracies than pruning after training (Liu et al., 2018). However, pruning an already trained network is often followed by fine-tuning effectively training the network twice (Fig. 3a). For comparability with pruning at initialization, we unify the overall training schedule between these two settings and consequently apply two training cycles after pruning the randomly initialized network (Fig. 3b; for further discussion, see App. A.5).

We observe that applying SOSP at initialization performs almost equally well than applying SOSP after training (see ResNet-56 and VGG in Fig. 3c and f, respectively). In conclusion, applying SOSP at initialization can significantly reduce the time and resources required for network training with no or only minor degradation in accuracy.

### 3.4 Identifying & Removing Architectural Bottlenecks

Even though the previous section highlighted that applying SOSP after training and at initialization results in comparable accuracies, the pruning masks differ between these two scenarios (compare Fig.

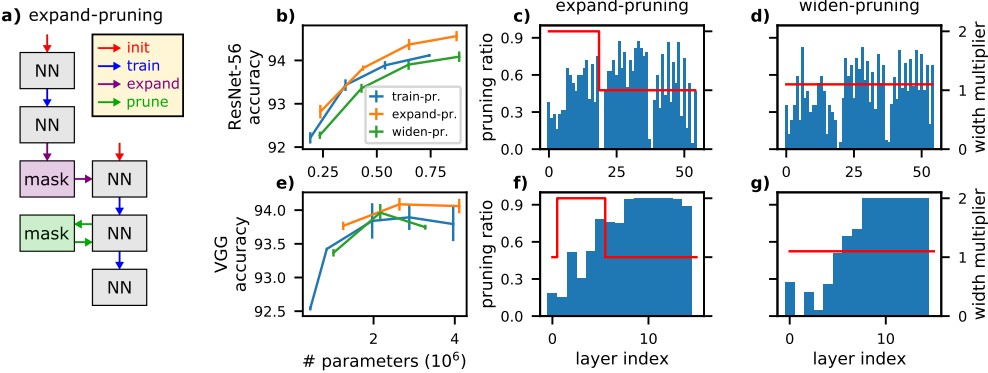

Figure 4: We remove architectural bottlenecks found by SOSP using the expand-pruning scheme (a) on Cifar10. The width of blocks and layers with low pruning ratios in the train-pruning scheme (Fig. 3d and g) are expanded by a width multiplier of 2 (c, f). As a baseline, we uniformly expand all layers in the network by a factor 1.1 (d, g). The layer-wise pruning ratios of the enlarged network models are shown as bar plots in (c, d, f, g). The average and standard deviation of the test accuracy across 3 trials are shown over the number of model parameters (b, e). Note that the full ResNet-56 and VGG models have $0.86 \cdot 10^6$ and $20 \cdot 10^6$ parameters, respectively.

3d and g to e and h, resp.). Despite this difference a common feature of all masks is that some layers are barely pruned or not pruned at all while others are pruned by up to $80\%$. This could indicate towards architectural bottlenecks. We consider a layer an architectural bottleneck if the respective layer has a considerably lower pruning ratio compared to the other layers. The low pruning ratio of bottleneck layers indicates that the substructures (e.g. filters) have a high sensitivity. Thus, widening these layers could improve the overall performance and allow for even smaller models with higher accuracies.

To utilize this insight we device a procedure that we call *expand-pruning*. The idea is to first calculate the pruning ratios of a trained network and then to identify architectural bottlenecks, i.e. the layers with the lowest pruning ratios. Next, we widen these layers by a factor of two, which has been empirically shown to work well. Finally, we randomly initialize, train, prune, and fine-tune the expanded network (for a schematic, see Fig. 4a). As a naive baseline that we call the *widen-pruning* procedure, we widen all layers in the network with a constant factor instead of widening specific layers. We choose the constant factor such that the overall number of parameters matches that of the expand-prune procedure (see width multiplier in Fig. 4).

We evaluate the expand-pruning procedure for ResNet-56 and VGG on Cifar10, for which we expand the least pruned of the three main building blocks and the five least pruned layers, respectively (e.g., see *width multipliers* in Fig. 4c and f selected on the basis of the pruning masks shown in Fig. 3d and g, resp.). Note that for ResNet-56 a more fine-grained removal of bottlenecks, e.g. on layer level, is not possible without changing the overall ResNet architecture. In summary, a selective removal of bottlenecks results in smaller network models with higher accuracy than pruning the vanilla network or unselectively increasing the network size (compare expand-pruning to train-pruning and widen-pruning in Fig. 4b and e). While in principle any global pruning method could be used for the expand-pruning procedure, SOSP is especially suited since it does not require to modify the network architecture like EigenDamage and can also be applied at initialization unlike NN Slimming, allowing for a similar expand scheme directly at initialization (see App. A.6).

## 4 Discussion

In this work we have demonstrated the effectiveness and scalability of our second-order structured pruning algorithms (SOSP). While both algorithms perform similarly well, SOSP-H is more easily scalable to large scale networks and datasets. We highlighted two major features of our method. Firstly, SOSP can be applied at initialization with only minor degradation in accuracy, which drastically reduces the required time and resources for training. Secondly, we showed that the pruning masks

found by SOSP can be used to systematically detect and remove architectural bottlenecks, further improving the performance of pruned networks.

Compared to other global pruning methods, SOSP captures correlations between structures by a simple, effective and scalable algorithm that neither requires to modify the training nor the architecture of the to be pruned network model and achieves comparable or better accuracies on benchmark datasets. The C-OBD algorithm (Wang et al., 2019a) is a structured generalization of the original unstructured OBD algorithm (LeCun et al., 1990). In contrast to OBD, C-OBD accounts for correlations within each structure, but does not capture correlations between different structures within and across layers. We show that considering these *global* correlations consistently improve the performance, especially for large pruning ratios (Tab. 1). This observation is further confirmed by an ablation study in which we neglect all cross-structure correlations significantly decreasing the performance under otherwise identical experimental settings (App. A.2). The objective of EigenDamage (Wang et al., 2019a) to include second order correlations is similar to ours, but the approaches are significantly different. EigenDamage uses the Fisher-approximation, which is similar to the Gauss-Newton approximation used for SOSP-I, and then, in addition to further approximations, apply low rank approximations that require the substitution of each layer by a bottleneck-block structure. Our SOSP method is simpler, easier to implement and does not require to modify the network architecture, but nevertheless performs on par with EigenDamage. The approach of NN Slimming (Liu et al., 2017) is more heuristic than SOSP and is easy to implement. However, networks need to be pretrained with $L_1$ regularization on the batch-normalization parameters, otherwise the performance is severly harmed (Tab. 1 and 3). SOSP does not require any modifications to the network training and can be applied to any pretrained network. A recent variant of NN Slimming was developed by Zhuang et al. (2020) who optimize their hyperparameters to reduce the number of MACs. Using the number MACs as an objective for SOSP is left for future studies.

In addition to the above comparison to other global pruning methods, we also compared our methods to simpler local pruning methods that keep the pruning ratios constant for each layer and, consequently, scale well to large-scale datasets. The pruning method closest to our SOSP-I method is the one by Peng et al. (2019). While both works consider second-order correlations between structures, theirs is based on a pruning objective different from our absolute sensitivites in $\lambda_2^I$ and considers only intra-layer correlations. Furthermore, they employ an auxiliary classifier with a hyperparameter, yielding accuracy improvements that are difficult to disentangle from the effect of second-order pruning. Going beyond a constant pruning ratio for each layer Su et al. (2020) discovered, for ResNets, that pruning at initialization seems to preferentially prune initial layers and thus proposed a pruning scheme based on a "keep-ratio" per layer which increases with the depth of the network. Our experiments confirm some of the findings of Su et al. (2020), but we also show that the specific network architectures found by pruning can drastically vary between different networks and especially between initialization and after training (histograms in Fig. 3). While all local pruning methods specify pruning ratios for each layer, our method performs automatic selection across layers (histograms in Fig. 3).

This automatic selection allows us to identify and remove architectural bottlenecks. However, our global pruning method has a bias towards pruning small structures, absent from local pruning methods, as the size of structures is usually identical within layers. We propose a simple solution by scaling each structure by the inverse of the kernel size which helps to remove some of the bias. Alternatively, to better reflect the computational costs in real-world applications, each structure could also be normalized by the number of its required MACs (like done by van Amersfoort et al., 2020).

Recently, unstructured (Lee et al., 2018; Wang et al., 2019b; Tanaka et al., 2020) and structured (van Amersfoort et al., 2020; Hayou et al., 2021) pruning schemes that are applicable to networks at initialization were proposed. While these methods fail to achieve similar accuracies compared to pruning after training, our SOSP method in the init-pruning setting achieves accuracies comparable to pruning after training.

In accordance with Elsken et al. (2019), our results suggest that pruning can be used to optimize the architectural hyperparameters of established networks (Liu et al., 2018) or super-graphs (Noy et al., 2020). Instead of formulating this optimization as a pruning process, we envision our second-order sensitivity analysis to be a valuable tool to identify and remove bottlenecks to find good neural architectures more quickly. For example, whenever a building block of the neural network cannot be compressed, this building block may be considered a bottleneck of the architecture and could be inflated to improve the overall trade-off between accuracy and computational cost.

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
