# OpenReview forum: "SOSP: Efficiently Capturing Global Correlations by Second-Order Structured Pruning"
_NeurIPS.cc/2021/Conference — NeurIPS 2021 Submitted_

### Official Review · Reviewer_mjMX · 2021-07-12

**Rating:** 6
**Confidence:** 4

**Summary:**

This paper presents two novel approximation techniques for approximating the importance of a filter (the estimated loss difference between pruned and pre-trained networks) using second order approximations. The main difficulties of second order approximation lie in the computational burden of computing the full Hessian matrix. To circumvent this issue, this paper presents two novel ways to do so. The first approach, dubbed SOSP-I, linearize the neural network and only consider the coupling of the loss function. The second approach, dubbed SOSP-H, approximate the loss difference using a Hessian-vector product. Both approaches have much better efficiency compared to using the Hessian and empirically perform a little better than existing approaches. Overall, the approximations seem novel to me and I find the appendix helpful in understanding the derivations. However, their empirical advantages seem rather limited. Additionally, I think some of the related papers are missed and they need to be discussed and compared against.

**Limitations And Societal Impact:**

I do not think this paper has negative societal impacts

**Main Review:**

## Strengths

- Novel approaches for approximating the Hessian for filter pruning
- The manuscript is well-written and easy to follow
- Filter pruning is a practically relevant research direction
- Comparisons with EigenDamage, NN-Slimming, and C-OBD are comprehensive for CIFAR

## Weaknesses

- The improvement over direct competitors (EigenDamage, NN-Slimming, and C-OBD) is relatively limited. Specifically, SOSP-H achieves 0.4% improvements over EigenDamage with similar MAC reduction for VGG on CIFAR-10 while being worse on CIFAR-100. Compared to EigenDamage, it seems only ResNet32 for CIFAR-100 works well. Not sure how to interpret this and an immediate question from the readers would be: When should we use EigenDamage and when should we use the proposed method? While I appreciate the results on ImageNet, direct competitors are not shown on ImageNet. More specifically, I do not think it is informative to compare local pruning methods with global pruning methods as the resulting pruned architectures are very important. It would be great if global pruning methods (e.g., EigenDamage, NN-Slimming, and C-OBD) are shown in the ImageNet table.

- Missing global pruning methods. Specifically, both [1,2] approximate the loss differences using Taylor expansion as well. Although missing second order information, I think it is not clear how much second order approximation adds to the picture if these first-order methods are not compared. Additionally, as noted by the authors in line 215, different layers contribute to MAC differently while the approximation only considers loss but not MAC. Hence, I think it is also critical to compared to learned global ranking such as [3] which factors the heterogeneous MAC into the ranking learning process.

- Missing comparisons with local pruning methods that have learned layerwise pruning ratio [4].

- Missing analysis on compact networks such as MobileNets and EfficientNets. Accelerating networks is important, but if the proposed method does not perform well on already fast/small networks, it is less significant.

[1] Theis, Lucas, et al. "Faster gaze prediction with dense networks and fisher pruning." arXiv preprint arXiv:1801.05787 (2018).

[2] Molchanov, Pavlo, et al. "Importance estimation for neural network pruning." Proceedings of the IEEE/CVF Conference on Computer Vision and Pattern Recognition. 2019.

[3] Chin, Ting-Wu, et al. "Towards efficient model compression via learned global ranking." Proceedings of the IEEE/CVF Conference on Computer Vision and Pattern Recognition. 2020.

[4] He, Yihui, et al. "Amc: Automl for model compression and acceleration on mobile devices." Proceedings of the European conference on computer vision (ECCV). 2018.

**Time Spent Reviewing:**

3

---

> ### Author Response · Authors · 2021-08-10
> **Response to Rev 4 (mjMX)**
>
> We first want to thank the reviewer for the extensive review, the positive feedback, and the great references provided. We believe the changes prompted by the reviewer really improved our paper. We address the individual comments in the following:
>
> 1.) Regarding the improvements of SOSP on CIFAR compared to previous global pruning methods, we kindly refer to the response to all reviewers. Regarding the question when to use which second-order method (e.g. EigenDamage, SOSP-I, SOSP-H, C-OBD),we generally recommend to use SOSP-H, since it outperforms or performs on par with all other methods while having the same low complexity as first-order methods. This is especially crucial for large-scale networks and datasets such as ImageNet, where to the best of our knowledge we present the first global second-order pruning method that scales to such sizes. Due to the high computational complexities, we could not include any results of EigenDamage or SOSP-I on ImageNet (see runtime of SOSP-I on ResNet-50 in reply to Reviewer 2 (edNm)). We have however included a further comparison to the global first-order method Taylor on ImageNet. Does this answer the comment raised by the reviewer?
>
> 2.) We are very thankful for the great references provided by the reviewer that we now discuss in the manuscript. Since [1] employs the Fisher approximation also used by EigenDamage which is also very similar to the approximation (6) used by SOSP-I, it will suffer from the same scaling behavior as SOSP-I (see runtimes referred to above). The reference [2] complements the findings of our paper very well and further provides global structured pruning results for ResNet-50 on ImageNet. We have included the results of Taylor [2] into our comparison (Table 2), showing quite clear advantages of SOSP:
>
> | ResNet-50 | MACs (10^9) | Params (10^7) | Acc (%) |
> |-----------|-----------|---------------|---------|
> | Taylor    | 2.66      | 1.79          | 75.48   |
> | SOSP      | 2.79      | 1.79          | 76.60   |
> | Taylor    | 1.34      | 0.79          | 71.69   |
> | SOSP      | 1.58      | 0.99          | 73.38   |
>
> It is mentioned in [2] that second-order pruning methods (only the diagonal of the Hessian or the OBD approximation is considered) tend to perform slightly better than first-order methods, but are usually also more costly. Therefore, reference [2] argues to use Taylor, a first-order pruning method, for large-scale networks and datasets. In contrast, SOSP additionally takes into account an approximation of the full second-order terms including the off-diagonal entries of the Hessian, but still scales as favorably as first-order methods. Further, SOSP outperforms both Taylor (see table above) and C-OBD (C-OBD disregards off-diagonal Hessian entries, see also Fig. 5 in our Appendix). Consequently, we conclude that while including some second-order information improves the results [2], including the full second-order terms further boosts the pruning performance. We thank the reviewer for the great references and hope for the reviewer’s agreement that the additional comparisons to Taylor [2] and the discussion of these two references strengthen our paper.
>
> 3.) We fully agree on the importance to take into account that different layers contribute differently to the overall MAC count. While the objective of our SOSP algorithm is to optimize the parameter or structure count, we also introduce a weighting of layers to reflect the MAC count in the objective of SOSP (Sec. 3.2, lines 211-217, and Tab. 2). In principal, it is also possible to use the actual MAC instead of the structure count as objective. Another method to include the MAC count into the optimization objective is proposed by [3] that we could adapt by substituting their L2-norm with our second-order approximation. A direct comparison of SOSP with the results of [3] is difficult because [3] uses, similar to CCP, the alternative and more optimistic MAC count (see App. D). Accounting for this difference in MAC counting (for ResNet-56 on CIFAR10, an approx. 50% MAC reduction in [3] corresponds to an approx. 40% MAC reduction with our more realistic MAC count) SOSP achieves an accuracy of 93.71% (see App. Tab. 5) vs. 93.7% for [3]. This necessarily rough comparison indicates that SOSP is on par with [3], even without using the number of MACs in the objective and without the tuning of hyperparameter.
>
> 4.) We again agree with the reviewer that a comparison to a local pruning method that learns layer-wise pruning ratios could be relevant. The finding that correlations between structures in different layers are important [4] confirms our results and further supports the need for an efficient way to calculate correlations between structures across layers, which our method achieves. While [4] significantly outperforms human-expert baselines, SOSP achieves significantly higher accuracy than [4]: For example, for an approx. 50% MAC reduction, ref. [4] achieves an accuracy of 91.9% vs. 93.27% of SOSP for ResNet-56 on CIFAR10 (see App. Tab. 5). We added this comparison to our discussion.

---

> > ### Comment · Reviewer_mjMX · 2021-08-15
> > **Responses to the rebuttal**
> >
> > I appreciate the authors' efforts in putting up a great rebuttal. In particular, I agree with the arguments regarding [3] that the proposed work can potentially be combined with [3] and is on par. Also, I agree comparing with first-order global pruning can greatly strengthen this work. However, the presented comparisons to Taylor in the Table and to [4] are not really apples to apples. Since the training hyperparameters can be very different, it is not clear if the improved performance comes from better training hyperparameters or better pruning. More specifically, I expect to see the following comparisons to be convinced by the arguments:
> >
> > - Compare with Taylor by replacing your approximation with theirs without any other changes. If the proposed method is indeed helpful due to a better approximation, we should observe higher pruned accuracy before and after fine-tuning.
> >
> > - Compare with learned layer-wise methods such as [4] by taking their learned architecture (the layerwise pruning ratios) and pruned your trained network accordingly based on filter norms, and further fine-tune it with your training pipeline. This tells us whether the proposed method really outperforms learned layerwise pruning ratios.
> >
> > I think the apples-to-apples comparisons with Taylor approximation are very important as the argument here is really better approximation with not so much overhead. Taylor has a really low overhead as it is first-order. Hence, a fair comparison with it can really help set up the stage for the proposed work.
> >
> > Additionally, it would be good to disentangle the performance achieved by the proposed method in terms of architecture and weights. According to the rethinking the value of pruning paper, it would be informative to help the reader understands if the proposed SOSP finds better architecture or weights, or both.

---

> > > ### Author Response · Authors · 2021-08-19
> > > **Response to Rev 4 (mjMX)**
> > >
> > > We thank reviewer 4 for the response and proposed experiments. As suggested by the reviewer, we performed several additional experiments.
> > >
> > > 1.)  We agree with the reviewer that an apples to apples comparison sheds more light on the role of the second-order terms for global pruning. Therefore, we conducted additional experiments comparing the objective of Taylor (first-order) with our SOSP-H objective for identical settings (only replacing the pruning objective). We provide results for ResNet-56 on Cifar-10/100 (see tables bellow) and triggered further experiments for other datasets and networks. The results show clearly that using second-order information improves the pruning performance consistently. In conclusion, SOSP- H scales similarly to Taylor, while outperforming Taylor.
> > >
> > > CIFAR10:
> > >
> > > | ResNet-56 (structured pruning ratio) | MACs (10^7) | Params (10^3) | Acc (%)    |
> > > |--------------------------------------|-------------|---------------|------------|
> > > | Taylor (0.1)                         | 108         | 739           | 93.90±0.13 |
> > > | SOSP (0.1)                           | 109         | 734           | 94.01±0.05 |
> > > | Taylor (0.3)                         | 76          | 531           | 93.44±0.19 |
> > > | SOSP (0.3)                           | 79          | 518           | 93.62±0.13 |
> > > | Taylor (0.5)                         | 50          | 356           | 92.98±0.09 |
> > > | SOSP (0.5)                           | 55          | 338           | 93.15±0.15 |
> > > | Taylor (0.7)                         | 28          | 195           | 91.55±0.29 |
> > > | SOSP (0.7)                           | 31          | 183           | 91.90±0.13 |
> > >
> > > CIFAR100:
> > >
> > > | ResNet-56 (structured pruning ratio) | MACs (10^7) | Params (10^3) | Acc (%)     |
> > > |--------------------------------------|-------------|---------------|-------------|
> > > | Taylor (0.1)                         | 105         | 750           | 71.18±0.10  |
> > > | SOSP (0.1)                           | 107         | 742           | 71.76±0.39  |
> > > | Taylor (0.3)                         | 71          | 562           | 71.11±0.07  |
> > > | SOSP (0.3)                           | 77          | 532           | 70.83±0.66  |
> > > | Taylor (0.5)                         | 45          | 388           | 68.89±0.09  |
> > > | SOSP (0.5)                           | 51          | 358           | 69.52±0.03  |
> > > | Taylor (0.7)                         | 23          | 291           | 65.51±0.18  |
> > > | SOSP (0.7)                           | 29          | 200           | 66.67±0.28  |
> > >
> > >
> > >
> > > 2.)  As proposed by the reviewer we investigated the effect of the architecture and weights. We compare the results of SOSP with Shuffle SOSP, where the pruning masks of the structures within each layer, and therefore also the weights, are shuffled before the fine-tuning step. Therefore, Shuffle SOSP retains the same architecture (the same number of structures are pruned per layer) as SOSP, but prunes different structures within each layer. The results for ResNet-56 on CIFAR-10 are shown in the table below. They suggest that the predominant factor is the architecture. While there seems to be a slight drop in performance comparing Shuffle SOSP to SOSP, this drop is generally smaller or similar to the drop from a second-order to a first-order pruning method.
> > >
> > > | Structured Pruning Ratio | SOSP       | Shuffle SOSP | Taylor     |
> > > |--------------------------|------------|--------------|------------|
> > > | 0.3                      | 93.62±0.13 | 93.59±0.12   | 93.44±0.19 |
> > > | 0.5                      | 93.15±0.15 | 92.94±0.14   | 92.98±0.09 |
> > >
> > >
> > >
> > >
> > >
> > > We added all results shown above to our paper and thank the reviewer for the suggestions.
> > > Do the additional results answer your questions?

---

> > > > ### Comment · Reviewer_mjMX · 2021-08-21
> > > > **Reply**
> > > >
> > > > Regarding comparing Taylor to SOSP, how's the performance after pruning "but before fine-tuning"? In my opinion, this has to be higher for SOSP compared to Taylor, and is it really the case? Lastly, it would be great to compare to methods that learn layer-wise ratios including [4] and actually [5,6,7]. This positions global pruning methods better compared to methods that try to learn layerwise pruning ratios. The authors can do so fairly by taking the learned layerwise ratios and train them using the authors' setup.
> > > >
> > > > Regardless, I think the additional experiments have partially addressed my concern and I am feeling comfortable bumping up the score by 1.
> > > >
> > > > [5] Guo, Shaopeng, et al. "Dmcp: Differentiable markov channel pruning for neural networks." Proceedings of the IEEE/CVF Conference on Computer Vision and Pattern Recognition. 2020.
> > > > [6] Li, Bailin, et al. "Eagleeye: Fast sub-net evaluation for efficient neural network pruning." European Conference on Computer Vision. Springer, Cham, 2020.
> > > > [7] Su, Xiu, et al. "Locally free weight sharing for network width search." ICLR 2021.

---

> > > > > ### Author Response · Authors · 2021-08-26
> > > > > **Response**
> > > > >
> > > > > We thank the reviewer for the suggestion to compare the accuracies of Taylor and SOSP before the fine-tuning step. The comparison is indeed insightful and confirms the reviewer's intuition that, even directly after pruning, SOSP should have higher accuracy than the first-order method Taylor. We show the accuracies of the pruned networks at epoch 0 of the fine-tuning step below. One can clearly see that the difference in performance between SOSP and Taylor grows significantly with increasing pruning ratios. The difference is even more striking than after fine-tuning (which was already given in the table in our previous response).
> > > > >
> > > > > CIFAR10:
> > > > >
> > > > > | ResNet-56 (struc. pruning ratio) | MACs (10^7) | Params (10^3) | Acc before fine-tuning (%) |Acc after fine-tuning (%)   |
> > > > > |--------------------------------------|-------------|---------------|------------|-----------|
> > > > > | Taylor (0.1)                         | 108         | 739           | 89.03±0.09 | 93.90±0.13 |
> > > > > | SOSP (0.1)                           | 109         | 734           | 88.81±0.62 |94.01±0.05 |
> > > > > | Taylor (0.3)                         | 76          | 531           | 86.19±0.48 |93.44±0.19 |
> > > > > | SOSP (0.3)                           | 79          | 518           | 86.88±0.31 | 93.62±0.13 |
> > > > > | Taylor (0.5)                         | 50          | 356           | 82.27±1.46 |92.98±0.09 |
> > > > > | SOSP (0.5)                           | 55          | 338           | 84.28±0.65 |93.15±0.15 |
> > > > > | Taylor (0.7)                         | 28          | 195           | 72.92±3.25 |91.55±0.29 |
> > > > > | SOSP (0.7)                           | 31          | 183           | 79.35±0.28 |91.90±0.13 |
> > > > >
> > > > > CIFAR100:
> > > > >
> > > > > | ResNet-56 (struc. pruning ratio) | MACs (10^7) | Params (10^3) | Acc before fine-tuning (%) |Acc after fine-tuning (%)   |
> > > > > |--------------------------------------|-------------|---------------|-------------|-----------|
> > > > > | Taylor (0.1)                         | 105         | 750           | 62.27±0.40  |71.18±0.10  |
> > > > > | SOSP (0.1)                           | 107         | 742           | 62.19±0.24  |71.76±0.39  |
> > > > > | Taylor (0.3)                         | 71          | 562           | 57.88±0.70  |71.11±0.07  |
> > > > > | SOSP (0.3)                           | 77          | 532           | 58.25±0.46  |70.83±0.66  |
> > > > > | Taylor (0.5)                         | 45          | 388           | 46.24±1.54  |68.89±0.09  |
> > > > > | SOSP (0.5)                           | 51          | 358           | 51.53±0.75  |69.52±0.03  |
> > > > > | Taylor (0.7)                         | 23          | 291           | 22.96±6.16  |65.51±0.18  |
> > > > > | SOSP (0.7)                           | 29          | 200           | 40.71±0.91  |66.67±0.28  |
> > > > >
> > > > > Further, we also ran an apples-to-apples comparison to AMC [4], which learns layer-wise pruning ratios, on PlainNet-20 (the only network for which layerwise pruning ratios are given in [4] for the structured pruning case). As suggested by the reviewer we took their learned architecture (Fig. 2 in [4]), then pruned each layer of our trained network based on the 2-norm importance ranking, as also done by [4], and then trained it with our fine-tuning pipeline. The results (see table below) show that while SOSP and AMC perform on par when comparing them over MACs, SOSP outperforms AMC significantly when compared over parameters (which is the actual objective of SOSP).
> > > > >
> > > > > | PlainNet-20 (struc. pruning ratio) | MACs (10^6) | Params (10^3) | Acc (%)     |
> > > > > |--------------------------------------|-------------|---------------|-------------|
> > > > > | AMC                                     | 20.9         | 132           | 90.23±0.10|
> > > > > | SOSP (0.3)                           |       26.7      |      124          |90.93±0.17  |
> > > > > | SOSP (0.4)                           | 22.7         | 90           | 90.14±0.15      |
> > > > >
> > > > > While [4] also provides an architecture (layerwise pruning ratios) for ResNet-50 on ImageNet, this architecture is obtained for unstructured pruning (called “fine-grained pruning” in [4]) and thus is not transferrable to a structured architecture, which is topic of our work.
> > > > >
> > > > > We hope that with these additional results we have now fully addressed the reviewer’s comments and questions. We again want to thank the reviewer for the fruitful and constructive discussion that definitely helped us to improve our paper.

---

### Official Review · Reviewer_qQtX · 2021-07-15

**Rating:** 4
**Confidence:** 4

**Summary:**

This work proposes a structured pruning method that removes feature maps (channels) of convolutional neural networks without losing accuracy while increasing computational efficiency (MACs). The proposed method (SOSP-I/H) is based on a sensitivity based saliency and involves what they call global correlations through the second order partial derivatives with respect to different structures. SOSP shares a lot in common with previous works including OBD and C-OBD for the use of second order information except that they approximately incorporate all elements in the Hessian matrix. Then the paper evaluates the effectiveness of the proposed method for image classification task using VGG, ResNet, and Desnet on CIFAR and ImageNet datasets and achieve accuracies on par with previous works on both global and local (layerwise) pruning. The paper further presents that SOSP works on pruning at initialization and can improve further by performing expand-pruning based on skewed distribution of retained parameters over layers.


**Limitations And Societal Impact:**

- The paper often misses clear explanations of why their approximation should work as intended. For instance, from (1) to (2) they take absolute values individually by saying that it is to avoid cancellations (L95-L98) however they clearly measure saliency differently from each other, and presumably the resulting pruned structures will most likely look very different.
- The method is not evaluated clearly how efficient the proposed algorithm is compared to other approaches; the computational cost for a high pruning ratio seems to be estimated not quite tight or realistic.
- A clear limitation of SOSP is that the method does not compare very strongly or improves over other existing prior works. Only in a few cases of global pruning seems to improve very marginally.
- The evaluations and analysis performed seem a bit insufficient to support claims such as the argument that SOSP at initialization works on par with SOSP after training while performing training twice and claim against other pruning approaches at initialization.
- Generalizing and automating expand-pruning rather than widening by a factor of two would make the process of finding bottleneck appeal stronger.



**Main Review:**

Using second order information to locally approximate the changes in the loss function can be useful, and in fact it has been used in many prior works as mentioned in the paper.
As it is quite expensive to compute the entire Hessian accurately for a deep network with a large number of parameters, the paper proposes to approximate based on (6) which is using the Hessian vector products.
Here, they make a rather strong assumption on the wide network referring to the NTK regime and the local first order approximation (linearization) of the network computation.
The paper attempts to demonstrate the effectiveness of the proposed method on large-scale experiments using ImageNet.


**Time Spent Reviewing:**

6

---

> ### Author Response · Authors · 2021-08-10
> **Response to Rev 3 (qQtX)**
>
> We thank the reviewer very much for the valuable feedback, which will especially make our presentation and explanations clearer in several places. We are happy to discuss some important issues raised by the reviewer.
>
> First, we would like to clarify a potential misunderstanding in “Main Review: The approximation (6) is only used for SOSP-I, and does not involve any Hessian vector products. The Hessian vector product is exclusively used in our main method SOSP-H, where, in contrast to SOSP-I, the Hessian itself is not approximated at all. We revised and clarified this paragraph to prevent any future misunderstandings.
> We further would like to emphasize that the approximation (Equation 5 and 6) used for SOSP-I is a well-established approximation that is also used in other pruning papers, e.g. the popular unstructured pruning method OBS [1], and for second-order NN training [2]. In contrast to [1, 2], we use this approximation for global structured pruning. We agree with the referee that this NTK approximation was theoretically proven only for strongly overparameterized networks. However, in practice, this approximation was also shown to work well for modern large networks [3], especially at stationary points such as after training. We clarified the method section in this regard and thank you for pointing this out.
>
> Next, we address the comment of the reviewer:
>
> 1.) We have derived and thus attempted to motivate our method on a theoretical basis. We however admit that some decisions in the end are rather based on empirical observations, e.g. where exactly to put absolute value signs in (2). We tried out several combinations and found the objective (2) with absolute individual saliencies to overall perform best. This can potentially be explained by the neural network not being at an exact (local) minimum or at a saddle point. Further, at least the pruning histograms of the different versions look actually quite similar empirically (more similar than Fig. 3d vs. 3e). Does this suffice as a clarification to the reviewer?  We modified the manuscript to make this clearer and thank the reviewer for pointing this out.
>
> 2.) (a) To clarify the efficiency of our algorithms, first, we derive their computational complexity in Sec. 2.3 and show that SOSP-H has the same favorable scaling as typical first-order methods. Second, we empirically show in Fig. 2 that SOSP-H runs significantly faster than SOSP-I on ResNet-56. We have also included runtime data for additional architectures in our paper (for numbers, we kindly refer to our response to Reviewer 2 (edNm)). The efficiency of our SOSP-H is further highlighted by the fact that we were able to apply SOSP-H to large-scale networks on ImageNet – something that for most other accurate methods could not be demonstrated (EigenDamage, SOSP-I) and which is considered to be difficult [4].
> (b) Regarding the computational cost for high pruning ratios, we remark that this cost is estimated quite tightly. It is for lower pruning ratios that the importance ranking would not have to be fully computed , lowering the O(S^3) and O(S log S) terms in the complexity, and overall reducing the computational cost. Does this answer the reviewer’s comment? We are not quite sure what the reviewer means by “the estimate is not quite tight or realistic for high pruning ratios”.
>
> 3.)Regarding the comparison to existing global pruning methods, we kindly refer to our response to all reviewers on this point.
>
> 4.)The reviewer is correct that the application of SOSP at initialization uses a different training schedule (“double-train”) than other works about pruning at initialization and is thus not directly comparable to these other works. We introduced the double-train scheme (Fig. 3b) to enable a fair comparison with pruning after training in terms of the number of total training epochs (and ultimately, in terms of accuracy). We see our results as a first indicator that SOSP applied at initialization achieves a comparable accuracy than SOSP after training with the advantage that the total training cost is decreased due to the reduced network size after pruning.
>
> 5.) We thank the reviewer for the suggestion to generalize the expand-pruning procedure. First, to clarify a potential misunderstanding, the factor of two is not applied to the whole network but just to these parts identified as bottlenecks. The factor of two was chosen after empirically evaluating different factors. We found that larger factors do not further improve the performance, while smaller factors lead to lesser improvement. An iterative tweaking this factor for each neural network could indeed be automatized, albeit at higher computational cost. Does this clarify the concerns of the reviewer? We will add further clarifications to the manuscript.
>
> [1] B. Hassibi, D. Stork. Second order derivatives for network pruning: Optimal brain surgeon. Advances in Neural Information Processing Systems (NeurIPS), 1992.
>
> [2] A. Botev, H. Ritter, D. Barber. "Practical Gauss-Newton optimisation for deep learning." International Conference on Machine Learning (ICML), 2017.
>
> [3] Lee, Jaehoon, et al. "Wide neural networks of any depth evolve as linear models under gradient descent." Advances in neural information processing systems 32 2019.
>
> [4] Molchanov, Pavlo, et al. "Importance estimation for neural network pruning." Proceedings of the IEEE/CVF Conference on Computer Vision and Pattern Recognition. 2019.

---

> > ### Author Response · Authors · 2021-08-21
> > **Response to Rev 3 (qQtX)**
> >
> > Did our response address your comments? Please do not hesitate to ask further questions.

---

### Official Review · Reviewer_edNm · 2021-07-16

**Rating:** 7
**Confidence:** 2

**Summary:**

This work proposes a second-order structured pruning algorithms (SOSP), which can drastically reduce training time and further improve pruned network performance by removing architecture bottlenecks

**Limitations And Societal Impact:**

This paper does discuss its limitations.

Questions and suggestions:

What would be the scale limits to choose SOSP-I over SOSP-H? Can we have more experiment in performance comparison between those two as network scale increases?


It would be interesting to compare with more recent pruning works such as  [1,2].

[1] AutoCompress: An Automatic DNN Structured Pruning Framework for Ultra-High Compression Rates.
[2] EagleEye: Fast Sub-net Evaluation for Efficient Neural Network Pruning



**Main Review:**

This work is well-explained and easy to follow. This paper has very detailed background overview on the state-of-the-art pruning techniques and recent findings. This paper presents proposed method in good detail and complexity analysis. Extensive experiments on CIFARs with multiple architectures effectiveness of the proposed approach in pruning and further conducted experiments demonstrated its effectiveness in removing architectural bottlenecks.


**Time Spent Reviewing:**

3

---

> ### Author Response · Authors · 2021-08-10
> **Response to Rev 2 (edNm)**
>
> We are grateful for the positive comments and feedback provided by the reviewer. We reply to the reviewer’s questions and suggestions.
>
> 1.) Concerning the performance comparison between SOSP-I and SOSP-H, we generally recommend to use SOSP-H for networks of any size. The only regime where SOSP-I becomes a valuable alternative is for small pruning ratios and small- to medium-sized networks. In the regime of small pruning ratios, the SOSP-H approximation degrades while the SOSP-I approximation is independent of the pruning ratio (and the importance ranking becomes cheaper, see point 2.b in reply to Reviewer 3 (qQtx)). Since this regime is usually uninteresting for single-shot pruning, the major runtime advantages of SOSP-H lead to our recommendation. We agree with the reviewer that we did not address this point carefully enough, consequently added a new table with additional runtime comparisons between SOSP-I and SOSP-H to the appendix, and extended the discussion about runtimes and application scenarios in the main text. The results in the added table (see the following excerpt) support the scaling found in Sec. 2.3 and complement our experiments in Fig. 2.
>
> | Time needed for pruning step (without training) | ResNet-56 | ResNet-32 | DenseNet-40 | VGG      | ResNet-50 (ImageNet) |
> |-------------------------------------------------|-----------|-----------|-------------|----------|----------------------|
> | SOSP-H                                          | 50±2s     | 123±4s    | 67±2s       | 110±3s   | 40±2min              |
> | SOSP-I                                          | 926±26s   | 1355±38s  | 635±18s     | 1647±44s | >48h                 |
>
> 2.)Thank you for providing us with these two very interesting works. To ensure a fair comparison we compare our work mostly with other single-shot pruning methods, especially other first- and second-order methods. The two works also make use of single-shot pruning methods, which are applied repeatedly. Thus, an efficient and well performing single-short pruning algorithm, such as SOSP, is essential and could potentially be used as a replacement for the pruning steps in [1] and [2]. Furthermore, even though a direct comparison to [1] might put SOSP at a disadvantage, for VGG with an approximately 90% MAC reduction, SOSP achieves a top (max over three runs) accuracy of 92.96 versus 93.2 for AutoCompress. This is surprisingly close. We now discuss these two papers in our manuscript. Thank you again for bringing these papers to our attention.

---

### Official Review · Reviewer_pfs4 · 2021-07-16

**Rating:** 6
**Confidence:** 3

**Summary:**

This paper proposed a structured pruning method which include correlations among structures by second order information. The author also improved the computational efficiency by fast Hessian-vector products. The method is demonstrated to achieve comparable/better results than previous methods. And it can also be applied to pruning-at-initialization and reveal architecture bottlenecks.

**Limitations And Societal Impact:**

Yes

**Main Review:**

Strength:
1. The method is well-motivated. The author provided thorough derivations to explain the method.
2. The method can be applied both after-training and at-initialization.
3. The application of identifying architecture bottleneck and correspondingly further improve the architecture is interesting.

Weakness:
1. In Table 1, it seems that the method only achieves comparable results with previous methods. In some cases it is better but the improvements are also very marginal.
2. The compared methods didn't seem to be the most recent ones. Most of them are in 2019 and before. Could the author conduct some comparison with the most recent state-of-the-art?

**Time Spent Reviewing:**

2

---

> ### Author Response · Authors · 2021-08-10
> **Response to Rev 1 (pfs4)**
>
> We thank the reviewer for the positive comments and feedback. We reply to the two points in question.
>
> 1.) The comparison to previous pruning methods in Table 1 is discussed in a common answer to all reviewers, to which we kindly refer the reviewer.
>
> 2.) We had scanned the recent conferences until the submission date for works on structured single-shot pruning methods, which are comparable to our method. While we indeed found more recent pruning methods, these methods employ iterative (not single-shot) pruning or include complex and expensive pipelines that make these methods hard to fairly compare to our single-shot SOSP method (see also point 2 in our response to Reviewer 2 (edNm)). Actually, our SOSP method could substitute for other single-shot methods used within some of these complex pruning pipelines. The newest single-shot pruning method we compare to is from 2020 (HRank) that we do not consider to be outdated. Do you have any specific references in mind that we missed to address?

---

> > ### Comment · Reviewer_pfs4 · 2021-08-31
> > **Post rebuttal**
> >
> > The author addressed my concerns. I will keep my score. However, I don't have a very high confidence in my assessment.

---

### Author Response · Authors · 2021-08-10
**Response to all Reviewers**

Several reviewers criticized that SOSP yields only minor accuracy improvements over other (global) pruning methods. We fully agree with the reviewers that this is the case for CIFAR10/100 in comparison to the global pruning methods NN-Slimming and EigenDamage, except for ResNet32 on CIFAR100, where SOSP significantly outperforms these approaches. However, SOSP outperforms C-OBD on two CIFAR architectures.

We would like to emphasize that the main advantage of our SOSP method is not only its accuracy, but also its scalability to large-scale networks and datasets. The computational complexity of our second-order method SOSP-H scales only like that of first-order methods (cf. Sec. 2.3), while still performing on par with or better than other second-order methods (e.g. EigenDamage, SOSP-I, FisherPruning). This allows us to apply SOSP-H to large-scale datasets (ImageNet) and networks – something that for the other methods could not be demonstrated. On ImageNet, SOSP outperforms all other local pruning methods, for which scalability is easier to achieve. Further, SOSP clearly outperforms global first-order methods (Taylor [cf. table in response to Reviewer 4 (mjMX)], which has the same complexity as SOSP). Additionally, SOSP is hyperparameter-free that further reduces the runtime compared to methods for which hyperparameters have to be optimized.

While NN-Slimming also features a low-complexity pruning step, NN-Slimming cannot be applied to neural networks with standard pretraining, but requires to train the neural networks with L1-regularization on the batch-norm parameters before pruning. Otherwise, the accuracy after pruning significantly drops, as shown in our Table 1. In contrast, SOSP can be applied right away to any neural network with standard pretraining and so remains efficient, as demonstrated in our ImageNet experiments. Further, SOSP can be applied at initialization and be used to remove architectural bottlenecks.

Summarizing, SOSP performs on par with or better than other pruning methods, despite its favorable first-order-like scaling to large networks and while being applicable to any pretrained network.

---

### Author Response · Authors · 2021-08-10
**Revision 1**

We thank all reviewers for the valuable feedback we have received. Since we are continuously updating our manuscript to incorporate the changes and additions suggested by the reviewers, we want to list here the changes to the manuscript already incorporated:

1.) In order to compare our results to a recent global pruning method and to demonstrate the advantage of using second-order information for global pruning, we added the global first-order pruning method “Taylor” to our ResNet-50 comparison on ImageNet. This follows a suggestion of Reviewer 4 (mjMX), please refer to our reply to this reviewer for details and results.

2.) We added a list with further runtime comparisons between SOSP-I and SOSP-H for several other architectures to the appendix of our paper (see our reply to Reviewer 2 (edNm) for details).

3.) We are now discussing the literature proposed by Reviewers 2 and 4 (edNm and mjMX; see also our replies to these reviewers).

4.) We incorporated several clarifications into our paper as suggested by the reviewers.

---

> ### Author Response · Authors · 2021-08-26
> **Revision 2**
>
> We thank especially Rev 4 (mjMX) for the valuable discussion, which helped us to further improve our manuscript. In addition to the changes of revision 1, we added the following experimental results suggested by Rev 4 to our paper:
>
> 1.)  A comparison of the first-order global structured pruning method Taylor within our pruning pipeline (changing only the pruning algorithm), before and after fine-tuning on ResNet-56 on Cifar10/100 (for results see our responses to Rev 4). We now also elaborate on the issue of first- vs. second-order methods in the motivational part of our paper (Introduction) and discuss it further in the Discussion section.
>
> 2.) A comparison of SOSP to AMC, which learns layer-wise pruning ratios by taking their learned architecture for PlainNet-20 and pruning our trained network based on the 2-norm within our pruning pipeline (for results see our response to Rev 4).
>
> 3.) We added a comparison of SOSP with Shuffle-SOSP (see response to Rev 4), indicating that the architecture (rather than the weights) are the predominant factor in structured pruning.

---

### Decision · Program_Chairs · 2021-09-27

**Decision:**

Reject

**Comment:**

This paper proposes a pruning technique that exploits second order information. In order to circumvent the high computational cost, an approximate computation is proposed. The improvements brought by the presented method are mostly marginal, and the method is not really compared to the most recent state of the art. Also, some of the critical parts of the paper lack clarity, for instance when motivating the approximation scheme. Overall, I recommend the rejection of this paper and encourage the authors to take the points raised by the reviewers into account when re-working the manuscript and preparing a submission to another venue.